# Isobolographic Analysis Demonstrates the Additive and Synergistic Effects of Gemcitabine Combined with Fucoidan in Uterine Sarcomas and Carcinosarcoma Cells

**DOI:** 10.3390/cancers12010107

**Published:** 2019-12-31

**Authors:** Marcin Bobiński, Karolina Okła, Jarogniew Łuszczki, Wiesława Bednarek, Anna Wawruszak, Gema Moreno-Bueno, Magdalena Dmoszyńska-Graniczka, Rafał Tarkowski, Jan Kotarski

**Affiliations:** 1I Chair and Department of Gynaecological Oncology and Gynaecology, Medical University of Lublin, 20-081 Lublin, Poland; 2I Chair and Department of Pathophisiology, Medical University of Lublin, 20-081 Lublin, Poland; 3Chair and Department of Biochemistry and Molecular Biology, Medical University of Lublin, 20-081 Lublin, Poland; 4Laboratorio de Investigación Traslacional, MD Anderson Cancer Centre Madrid, Calle de Arturo Soria, 270 28033 Madrid, Spain

**Keywords:** uterine sarcoma, gemcitabine, fucoidan, isobolography

## Abstract

Background: Uterine sarcomas and carcinosarcoma are associated with unfavorable prognosis. The regimens that are used in chemotherapy are associated with high incidence of side effects and usually do not significantly increase patients’ survival rates. In this study we investigated the activity and interactions between gemcitabine and fucoidan, the natural compound known for its anti-tumor properties, in human sarcomas and carcinosarcoma cell models. Methods: SK-UT-1, SK-UT1-B (carcinosarcoma), MES-SA (leiomyosarcoma), and ESS-1 (endometrial stromal sarcoma) cell lines were used for the experiments. Cells were incubated in the presence of gemcitabine, fucoidan, and mixtures, after the incubation the MTT tests were performed. In order to assess the interactions between tested compounds isobolographic analysis was performed. Additional assessments of apoptosis and cell cycle were done. Results: Additive effect of combined treatment with gemcitabine and fucoidan was observed in ESS-1 and SK-UT-1 cell line. Although the supra-additive (synergistic) effect noticed in SK-UT-1B cell line. It was not possible to determine the interactions of fucoidan and gemcitabine in MES-SA cell line due to insufficient response to treatment. Addition of fucoidan to gemcitabine enhances its proapoptotic activity, what was observed especially in ESS-1 and SK-UT-1B cell lines. The arrest of cell cycle induced by mixture of gemcitabine and fucoidan, superior comparing gemcitabine alone was observed in SK-UT-1B. Conclusions: Obtained data showed that a combination of fucoidan and gemcitabine in uterine endometrial stromal sarcoma and carcinosarcoma cell lines has additive or even synergistic effect in decreasing cell viability. Furthermore, this drug combination induces apoptosis and arrest of cell cycle. The resistance of uterine leiomyosarcoma cell line, justifies searching for other drugs combinations to improve therapy efficacy.

## 1. Introduction 

Uterine sarcomas are a group of malignancies consist of various types of tumors arising from mesenchymal tissue. The most common type of uterine sarcomas is leiomyosarcoma (about 60% of cases), less common types are endometrial stromal sarcoma (low and high grade), liposarcoma, rhabdomyosarcoma, and many other rare types [1]. The incidence of uterine sarcomas remains low respectively to epithelial malignancies (3–7% of all uterine malignancies) [2]. The diagnosis of uterine sarcoma is associated with bad prognosis and low rate of response to chemotherapy. Carciosarcoma is a type of tumor consisting of both mesenchymal and epithelial cells, that nowadays is considered as a type of endometrial cancer. These tumors are associated with bad prognosis and, similarly to uterine sarcomas, the results of its systemic treatment remain unsatisfactory. Systemic treatment of uterine mesenchymal tumors is based on cytostatic agents such as doxorubicine, gemcitabine, docetaxel, dacrbazine, and ifosfamde [3]. 

The exceptions from the above characteristics are endometrial stromal sarcomas, associated with indolent clinical behavior and favorable prognosis. The majority of these tumors express estrogen receptors and usually respond to hormonal treatment [2]. 

The retrospective trial comparing gemcitabine plus docetaxel, ifosfamide plus cisplatin, doxorubicin plus ifosfamide, ifosfamide alone, topotecan alone, and observation only in stage I and II leiomyosarcoma revealed no significant differences between groups in recurrence rate [4]. Due to the rarity of uterine sarcomas, the number of completed, randomized, prospective trials is limited. The response rates among patients suffering from uterine sarcomas, in prospective trials assessing the efficacy of various combinations of gemcitabine, docetaxel, and bevacizumab were between 25% and 35.8% [5,6,7].

The results obtained in systemic therapy of sarcomas are unsatisfactory. Most authors strongly recommend conducting clinical and pre-clinical trials in this field. The situation described above led to a search for new agents to be active against uterine sarcomas. Fucoidan is sulphated polysaccharide derived from brown seaweed, that recently gained attention due to its biological activities. It is known to affect multiple pathways in cancer cells including PI3K/AKT, MAPK, PTEN, VEGF, and caspases. Its effectiveness was proven in various models including lymphoma, leukemia, prostate cancer, breast cancer, hepatic cancer [8]. Recently we reported the anti-proliferative and pro-apoptotic activity of fucoidan in monotherapy among uterine sarcoma and carcinosarcoma cell lines. Simultaneously we confirmed previous observations that fucoidan do not affect normal (benign) human cells. Additionally, fucoidan is widely using as a dietary supplement. The characteristics listed above allow it to be considered as a safe product [8,9,10].

Obtained results led us to question the activity of fucoidan in combination with standard chemotherapy among uterine sarcomas. Gemcitabine was selected to be investigated together with fucoidan. Gemcitabine is commonly used anti-tumor drug belonging to the group of antimetabolites. It express antimetabolic effect by interruption of DNA synthesis. It is widely used as an option of standard approach for the systemic therapy in most of uterine sarcomas subtypes [3,11,12]. Furthermore, it is a comparator in many ongoing clinical trials enrolling uterine sarcoma patients [12]. Combinations of gemcitabine with novel agents including olaratumab, nivolumab, and pazopanib are under investigation in phases II–III of clinical trials [13].

This study was aimed to test if the combination of gemcitabine and fucoidan will have better therapeutic effect in uterine sarcomas and carcinosarcomas cell lines than the compounds applied alone and to assess the type of interactions between them in concomitant treatment. 

## 2. Materials and Methods

### 2.1. Reagents

Gemcitabine (100 mg/mL, 0.38 mMol/mL) was purchased from Accord Healthcare (UK), and fucoidan (*Undaria pinnatifida*) was obtained from Sigma-Aldrich (St. Louis, MO, USA). Fucoidan was diluted in respective complete culture medium at a concentration of 10 mg/mL just before use.

The Roswell Park Memorial Institute 1640 (RPMI-1640), Eagle’s Minimum Essential Medium (MEM), McCoy’s 5a Medium Modified, fetal bovine serum (FBS), trypsin-EDTA were purchased from PAN-Biotech (Aidenbach, Germany), penicillin-streptomycin and MTT (3-(4,5-dimethylthiazol-2-yl)-2,5-diphenyltetrazolium bromide) were obtained from Sigma-Aldrich (St. Louis, MO, USA). The Cell Proliferation ELISA System assay kit was purchased from Roche (Molecular Biochemicals, Manneihm, Germany). PE Active Caspase-3 Apoptosis Kit and Propidium iodide utilizing the PI/RNase Staining Buffer were obtained from Becton Dickinson Biosciences (San Jose, CA, USA). 

### 2.2. Cell Lines and Cultures

Carcinosarcoma cell lines (SK-UT-1, SK-UT1-B), uterine leiomyosarcoma cell line (MES-SA), and endometrial stromal sarcoma cell line (ESS-1) were obtained from the Laboratorio de Investigación Traslacional (MD Anderson Cancer Center, Madrid). The selection of cell lines was done in order to assess the differences in response to tested compounds among wide range of uterine sarcomas. Each cell line used for experiments is a model of particular tumor what makes obtained data more reproducible and comparable with other research. 

The cells were cultured in MEM (SK-UT-1, SK-UT1-B), McCoy’s 5a Medium Modified (MES-SA), RPMI-1640 (ESS-1), containing 10% (SK-UT-1, SK-UT-1B, MES-SA) or 20% (ESS-1) FBS and 1% penicillin-streptomycin at 37 °C in a humidified 5% CO_2_ atmosphere. Cells from the 4th to 9th passage were used for all experiments.

Detailed characteristic of cell lines is presented in Table 1.

### 2.3. Cell Viability Assay

Cells were platted on 96-well microplates SK-UT-1, SK-UT-1B, MES-SA, and ESS-1 (3 × 10^4^ cells/mL) and cells were incubated in the presence of gemcitabine (0.1–200 ng/mL), fucoidan (0.01–5 mg/mL) and mixtures of both compounds for 96 h. The maximum concentration of gemcitabine achieved in human serum was 26.79 ± 10.06 μg/mL, reported by Wang et al. so the concentrations used in our experiments were much lower comparing to ones available in vivo [16]. The safe concentrations of fucoidan in human serum are still under investigations and no conclusive data are available up to date. Afterwards, the cells were incubated for 3 h with the MTT. During the time MTT was metabolized by living cells to purple formazan crystals, which were later solubilized in SDS buffer (10% SDS in 0.01 N HCl) overnight. Separate experiments were performed in triplicate. The optical density of the product was measured at 570 nm with the use of an ELX-800 plate reader (Bio-Tek, Instruments, Winooski, VT, USA) and analyzed using Gen5 software (Bio-Tek, Instruments, Winooski, VT, USA).

### 2.4. Assessment of Apoptosis

Cell lines were treated with tested compounds and mixtures for 48 h, as a control cells without treatment were used. Afterwards cells were harvested, fixed and permeabilized using the Cytofix. All of experiments were performed according to the manufacturer’s instructions of PE Active Caspase-3 Apoptosis Kit (BD Biosciences, San Jose, CA, USA). Labeled cells were analyzed by flow cytometer FACSCalibur (Becton Dickinson, Franklin Lakes, NJ, USA), operating with CellQuest software (BD Biosciences, San Jose, CA, USA) to quantitatively assess the caspase-3 activity. The methodology of apoptosis assessment were described in detail elsewhere [9].

### 2.5. Cell Cycle Analysis

Cell lines were treated with tested compound and mixture for 48 h (as a control we used cells without exposure to tested compounds) and then fixed in 80% ethanol at −20 °C for 24 h. The experiment was conducted by utilizing PI/RNase Staining Buffer (BD Biosciences, USA) according to the manufacturer’s instructions. Cell cycle analysis was performed using flow cytometry (FACSCalibur (Becton Dickinson, USA). Acquisition rate was at least 60 events/sec in low acquisition mode and at least 10,000 events were measured. Methodology of cell cycle analysis was presented in detail previously [10]. 

### 2.6. Isobolographic and Statistical Analysis

In order to determine the inhibition rate of cell viability, measured by MTT assay, per dose of gemcitabine and fucoidan, log-probit linear regression analysis was performed according to method described by Litchfield and Wilcoxon [17,18]. Median inhibitory concentrations (IC_50_) for gemcitabine and fucoidan in ESS-1, SK-UT-1, and SK-UT-1B cell lines were calculated according to method previously described [19]. Due to lack of cells response, the IC_50_ was not achievable for fucoidan in MES-SA cell line [10]. Parallelism between dose–response curves for gemcitabine and fucoidan in ESS-1, SK-UT-1, and SK-UT-1B cell lines was confirmed by log-probit method, as it was described in detail previously [18]. Next, isobolographic analysis of interactions between drugs for the combination of gemcitabine and fucoidan in tested cell lines were performed according to the method presented by i.a. Tallarida et al. [20]. The median additive inhibitory concentrations (IC_50 add_) for two-drug mixtures were theoretically calculated according to the method described elsewhere [20]. The calculated values were used for performing MTT tests on ESS-1, SK-UT-1, and SK-UT-1B cell lines—the assessment of experimentally derived IC_50 mix_ values for tested drug combinations in a fixed 1:1 ratio. The particular drug concentrations (gemcitabine and fucoidan) in the mixture were calculated by multiplying IC_50 mix_ values accordingly to proportions in additive mixture. Detailed description of isobolographic method was introduced by Tallarida, Grabovsky and Luszczki [20,21,22]. The results of MTT test were analyzed by one-way ANOVA test, Tukey’s Multiple Comparison Post-test using GraphPad Prism 5.0 (GraphPad Softwere Inc., San Diego, CA, USA). The *p* < 0.05 was considered as statistically significant. 

## 3. Results

### 3.1. Cell Viability Assay 

Anti-proliferative effects of gemcitabine on tested cell lines is presented on Figure 1. Experimentally determined IC_50_ values for gemcitabine in SK-UT-1, SK-UT-1B, ESS-1, and MES-SA cell lines, were 31.173, 25.243, 13.875, and 72.482 ng/mL respectively. 

As we previously reported fucoidan significantly affects SK-UT-1, SK-UT-1B, and ESS-1 cell lines, meanwhile MES-SA cells seem to be resistant for this agent. IC_50_ was 0.966, 3.348, and 0.848 mg/mL respectively, it was not possible to determine IC_50_ for fucoidan in MES-SA cell line due to insufficient response to treatment [9]. The IC_50_ values are summarized in Appendix A.

### 3.2. Isobolographic Anaysis

Additive effect of the combined treatment with gemcitabine and fucoidan was observed in ESS-1 and SK-UT-1 cell lines. Although the supra-additive (synergistic) effect was noticed in SK-UT-1B cell line. 

The details of results obtained in isobolographic analysis are presented on Figure 2, Figure 3 and Figure 4.

In Figure 2, Figure 3 and Figure 4 the median inhibitory concentrations (IC_50_) for gemcitabine (GEM) and fucoidan (FUK) are plotted on the X- and Y-axes, respectively. The solid lines on both axes reflect the S.E.M. for the IC_50_ values for the studied drugs, when administered alone. The lower and upper isoboles of additivity represent the curves connecting the IC_50_ values for GEM and FUK administered alone. The dotted line illustrates the fixed-ratio of 1:1 for the combination of GEM with FUK. The points A’ and A” depict the theoretically calculated IC_50 add_ values for both, lower and upper isoboles of additivity. The point M reflects the experimentally-derived IC_50 mix_ value for total dose of the mixture expressed as proportions of GEM and FUK that produced a 50% anti-proliferative effect (50% isobole) in the cancer cell line (SK-UT-1, SK-UT-1B, and ESS-1, respectively for Figure 2, Figure 3 and Figure 4) measured in vitro by the MTT assay. On the graph, the S.E.M. values are presented as horizontal and vertical error bars for every IC_50_ value. Type I isobolographic analysis of interactions are presented in Appendix A. The effect of combined fucoidan and gemcitabine on the proliferation of tested cell lines were presented of Figure 5.

### 3.3. Assessment of Apoptosis

The impact of gemcitabine and its combination with fucoidan on apoptosis, measured as a number of cells with activated caspase 9, is presented on Figure 6A–C. Gemcitabine was used in concentration of IC_50_. Results obtained in isobolographic analysis were used to select appropriate concentrations of mixture for apoptosis assessment. For SK-UT-1 and SK-UT-1B concentrations of 0.5 IC50 for both agents were used. Due to very strong effect observed in ESS-1 cell line concentration 0.05 IC_50_ was selected for experiments.

The strongest induction of apoptosis by both gemcitabine and mixture gemcitabine + fucoidan was observed in ESS-1 cell line. This effect was also observed in SK-UT-1B cells. Although the induction of apoptosis in SK-UT-1 cell line of both single agent and combination was very weak. Statistical significance was observed between single agent and combination but the differences comparing to control were not significant. 

### 3.4. Cell Cycle

The results of cell cycle analysis of cells treated with gemcitabine and its mixture with fucoidan are presented in Figure 7. Statistical significance of differences between each phase of cell cycle in treatments and control are listed in Appendix A. Tested compounds were used in concentration as in apoptosis assessments. 

The most significant impact of concomitant treatment with gemcitabine and fucoidan on cell cycle arrest (measured as percent of cells in phases pre-G1 and G0/G1) was detected in the SK-UT-1 B cell line (comparing to both control and gemcitabine treatment). In the ESS-1 cell line significant differences were noted between cells treated with gemcitabine and control, with no such differences between gemcitabine and mixture. The differences in number of cells in each phase of cell cycle in SK-UT-1 cells exposed to investigated agents were very small in spite of statistical significance observed in particular phases. 

## 4. Discussion

This paper for our knowledge is the first report of interactions between fucoidan and gemcitabine in any model. The results obtained in our experiments show that both gemcitabine and fucoidan significantly affect cell viability in all tested cell lines (with exception of fucoidan in MES-SA cell line). The problem of MES-SA cell line resistance to fucoidan have been discussed previously [10]. In present study we observed also the worst response of MES-SA to gemcitabine among all of tested cell lines. Interestingly we used regular MES-SA line, but not its multidrug resistant variant MES-SA/Dx5. The mechanism of relative resistance of MES-SA cell line to fucoidan and gemcitabine is not known and requires further investigations. For our best knowledge the phenomenon of spontaneous MES-SA resistance to gemcitabine have not been published yet. Resistance to gemcitabine was artificially induced by genetic modifications (transfection) in gene coding deoxynucleoside kinase in MES-SA, although such modifications were not performed in the cell line we used [23]. Occurrence of some spontaneous mutations in deoxynucleoside kinase gene at least in part of cells could possibly explain our observation. The occurrence of serious side effects combined with limited activity makes systemic therapy of uterine sarcomas doubtful [13]. The most common unwanted results of chemotherapy are hematologic toxicities (leucopoenia, neutropenia, thrombocytopenia), fever, and concomitant infectious, that appear in the majority of treated patients [24]. Side effects of drugs are mostly dose-dependent, so it could be limited by the reduction of dosage, on the other hand, reduction of dose would decrease the therapeutic effect. 

In multidrug regimens, in assumption, lower doses of multiple drugs may cause fewer side effects (or its lower intensity) without decreasing effectiveness or even increasing it [13]. But in clinical practice such approaches are commonly associated with higher rate of adverse events. 

The rarity and heterogeneity of uterine sarcomas limit the possibilities of conducting clinical trials, which may be considered as a cause of the poor results of its treatment up to date. Even in preclinical studies number of research aimed to test new drugs combinations in this group of tumors is limited. 

Coley et al. tested the combination of seliciclib (cyclin-dependent kinases (CDK)-inhibitor) and paclitaxel among selection of uterine sarcomas cell lines. Using isobolography, they observed synergism between both drugs. Interestingly, differences in activity depending on the sequence of each drug administration were reported. Although the study is interesting, seliciclib is not registered for the treatment of uterine sarcomas. Furthermore, paclitaxel in monotherapy is not recommended among this type of malignancies [25].

In the present study we propose a combination of standard chemotherapy with a natural compound that is known to be safe; its intake is not associated with serious adverse events [26]. Both substances were administered at the same time in order to avoid a “drug sequence effect” observed by Coley at al. that could influence the results. Isobolographic analysis showed additive interaction between gemcitabine and fucoidan in SK-UT-1 and ESS-1 cell lines and even supra-additive (synergy) in SK-UT-1B. No antagonism between tested substances was observed. 

Cell viability was decreased equally or more than the sum of effects of single agents. If these observations are confirmed in animal models, clinical trials of fucoidan as an addition to treatment with gemcitabine may allow a decrease of its dose. Fucoidan is nowadays used as dietary supplement, and seaweeds containing it are widely used in Asian cuisine [26]. There are many researches aimed at checking its activity in various types of cancer. Some of them regard a combination of fucoidan and other agents (including cytostatics), and the results of most of them are promising. In our study we assessed the type of interactions between fucoidan and a cytostatic agent, using an isobolographic method, and calculated IC50 for the mixture. Significantly lower doses of therapeutic compounds may be associated with lower risk of adverse events. Furthermore, it may solve another problem that was widely discussed previously— the bioavailability of fucoidan. That is currently still under investigation both in animal models as well as in clinical trials—the results of these studies are awaited [27,28,29]. IC50 values for fucoidan obtained in combination with gemcitabine are much lower than 1 mg/mL: Such concentrations seem to be achievable in vivo. The doubtful concentrations were around 5 mg/mL. Although a few studies in this field were performed, precise determination of available serum concentrations among human are still under investigation [29,30].

Quite similar results were obtained by Mathew et al. who assessed IC50 for various types of fucoidan from 0.3 to 1.3 mg/mL (depending on plant that it was derived from). In this study they also performed analysis of fucoidans’ impact on CYP450 and COMT (catechol-O-methyltransferase) pathways, concluding that it is limited, which might be considered as a confirmation of its safety [31].

Our findings are comparable with other studies assessing activity of fucoidan and cytotoxic drugs. Burney et al. assessed activity of 2 types of fucoidan in combination with paclitaxel and tamoxifen on mouse models of human breast or ovarian cancer. They concluded that interaction between tested compounds were additive or supra-additive (depending on combination and model used) [32].

Similar observations were taken by Zhang et al. who tested combinations of fucoidan with cisplatin, tamoxifen, and paclitaxel in breast cancer cell lines. They concluded that fucoidan “enhances the anti-cancer activity” of tested cytostatic drugs. Furthermore, they observed the induction of apoptosis and cell cycle arrest among cells treated with fucoidan what corresponds with our previous findings [9,33].

Observed additive and supra-additive interactions between various fucoidans and chemotherapeutic drugs require explanation. The mechanism of fucoidans’ activity is not fully investigated. As it was mentioned above there are various cellular pathways that are affected by this compound. Its multipotential activity is, on the one hand, promising, and does not allow the overcoming of single pathway change, but on the other it is very difficult to understand completely. One of possible mechanism of synergistic interaction with chemotherapeutics is the ability to down-regulate expression of Bcl-xL and Mcl-1, known as anti-apoptotic proteins [34]. The addition of fucoidan to gemcitabine enhances its ability to induce apoptosis. This effect is especially noticeable in ESS-1 cell line where, even if mixture was added in concentration 0.05 of IC50, almost 80% of cells enter apoptosis and the difference between gemcitabine in monotherapy and the mixture reaches almost 20%. Weaker effect was observed in SK-UT-1B cells. In the SK-UT-1 cell line almost no impact of gemcitabine and fucoidan to apoptosis was noted, even if agents were used in concentration of 0.5 of IC50. Abudabbus et al. compared the ability of fucoidan in combination with cisplatin, doxorubicin and taxol, to induce apoptosis and to arrest the cell cycle among benign and malignant breast cells [35]. They reported a strong impact to cell cycle arrest and induction of apoptosis in malignant cells but no significant effect in normal cells. A comparable effect of fucoidan in monotherapy was described by Arumugam et al. in hepatoblastoma cell line [36]. So, our results confirm proapoptotic features of fucoidan in selected models. Interestingly, observations taken in experiments assessing apoptosis were confirmed also in cell cycle analysis. The strongest effect of combination of gemcitabine and fucoidan was observed in ESS-1 line. Moderate and weak changes in number of cells in particular cell cycle phases were observed in SK-UT-1 and SK-UT-1B cell lines respectively. 

Gemcitabine is known to affect the cell cycle in various cell models and in vivo; this feature is the consequence of its mechanism of action by DNA damaging [37]. The process of gemcitabine-induced cell cycle arrest can by interrupted by activity of DNA repairing mechanisms such as Chk1 [38]. Such a mechanism might be responsible for the limited changes in cells distribution among cell cycle phases observed in SK-UT-1 and SK-UT-1B treated with gemcitabine.

Park et al. reported the effect of cell cycle arrest by fucoidan treatment in bladder cancer cells. They also revealed the its association with down-regulation of cyclin D1, cyclin E, and cyclin-dependent-kinases (Cdks) in a concentration-dependent manner, without any change in Cdk inhibitors (p21, p27) [39]. Interestingly Han et al. experimented on a colon cancer model and observed cell cycle arrest associated with an increased expression of p21 [40]. Different mechanisms of cell cycle arrest induced by fucoidan in different models, combined with potential activity of DNA repairing factors affecting activity of gemcitabine, can explain differences among tested cell lines, we observed. So far, no studies assessing the impact of combination of gemcitabine and fucoidan have been published. 

The results and survey of literature indicate that testing multicompound regimens gives promising results and further research in the field could reveal a combination that could be beneficial for patients suffering from uterine sarcomas.

The study is not without limitations and should be considered carefully. The main limitation of the study is a cancer model that was used, 2D cell cultures, which do not allow investigation of the tumor microenvironment and may not fully reflect the response of tumor cells in vivo. Although promising, obtained results require clinical studies to be considered in practice.

The results obtained in our study indicate 3 potential directions for further investigations:Analysis of the mechanisms of the MES-SA cell line relative resistance to fucoidan and gemcitabine that may lead to identification of such mechanisms in tumors and make progress for overcoming them.Due to similarities in clinical course between carcinosarcomas and endometrial carcinomas the results obtained on SK-UT-1 and SK-UT-1B cell lines might be replicated on carcinomas cell lines and extend indications for future practical applications.Studies on 3D cell cultures or animal models have to be performed in order to confirm the activity of proposed combination on tissue models and to assess how the tumor microenvironment affects it.

## 5. Conclusions

Obtained data showed additive and supra-additive effect of fucoidan combined with gemcitabine in uterine endometrial stromal sarcoma (ESS-1) and carcinosarcoma (SK-UT-1, SK-UT-1B) cell lines, what confirms it has better or at least equal performance comparing to sum of effects of monotherapies. 

The addition of fucoidan to gemcitabine enhances proapoptotic effect of gemcitabine in endometrial stromal sarcoma cells (ESS-1) but not in carcinosarcoma (SK-UT-1, SK-UT-1B) cell lines.

Gemcitabine in monotherapy do not induce cell cycle arrest in carcinosarcoma (SK-UT-1, SK-UT-1B) cell lines. Although the addition of fucoidan to gemcitabine induces it in model of carcinomatous part of carcinosarcoma (SK-UT-1B).

Differences in response to applied treatment among tested cell lines (in cell viability apoptosis and cell cycle distribution) can be explained by the multipotential and not-fully-investigated activity of fucoidan as well as differences in cellular mechanisms (such as DNA repairing) in selected models.

The relative resistance of uterine leiomyosarcoma cell line (MES-SA) to applied drugs combination justify searching for other therapeutic regimens to improve therapy efficacy.

Taking into consideration the disappointingly low effectiveness of systemic therapy among these types of cancer, a combination of gemcitabine and fucoidan seems to be a promising alternative, having the potential to increase effectiveness and safety of the treatment. 

## Figures and Tables

**Figure 1 cancers-12-00107-f001:**
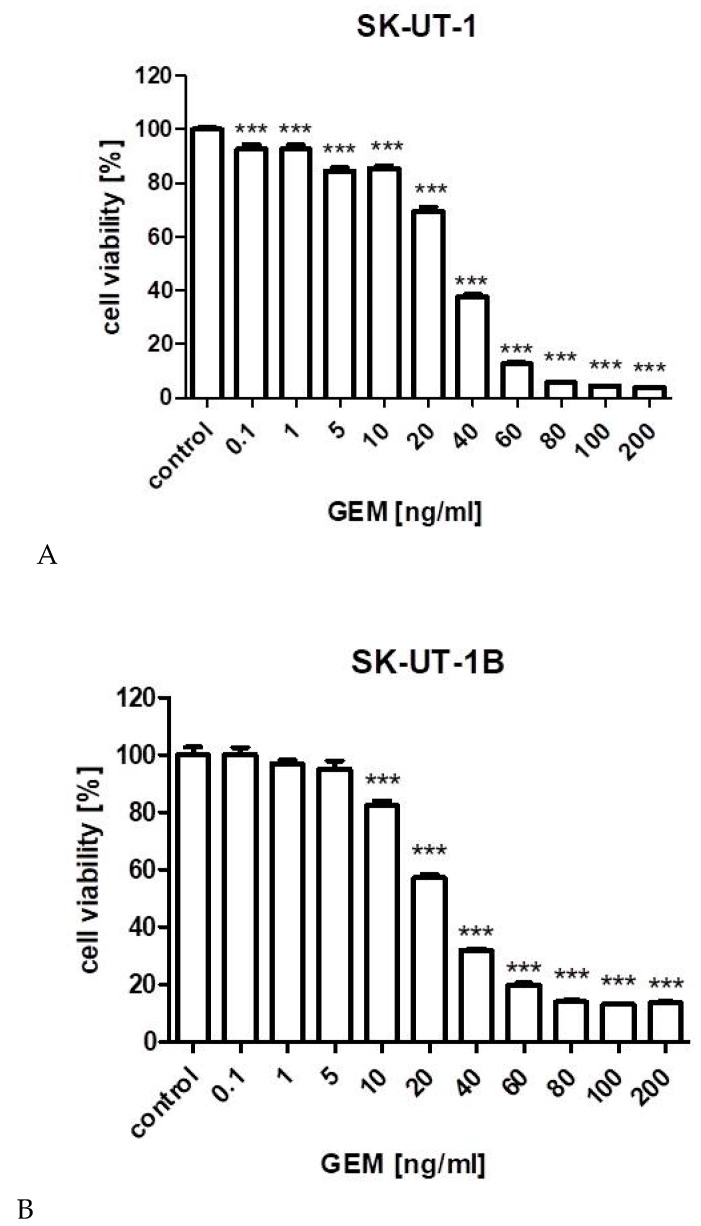
The influence of gemcitabine on the proliferation of carcinosarcoma cell lines (SK-UT-1 (**A**), SK-UT1-B (**B**)), endometrial stromal sarcoma cell line (ESS-1 (**C**)) and uterine leiomyosarcoma cell line (MES-SA (**D**)). The cells were treated with the gemcitabine at various concentrations for 96 h. (** *p* < 0.01, *** *p* < 0.001 were considered as statistically significant).

**Figure 2 cancers-12-00107-f002:**
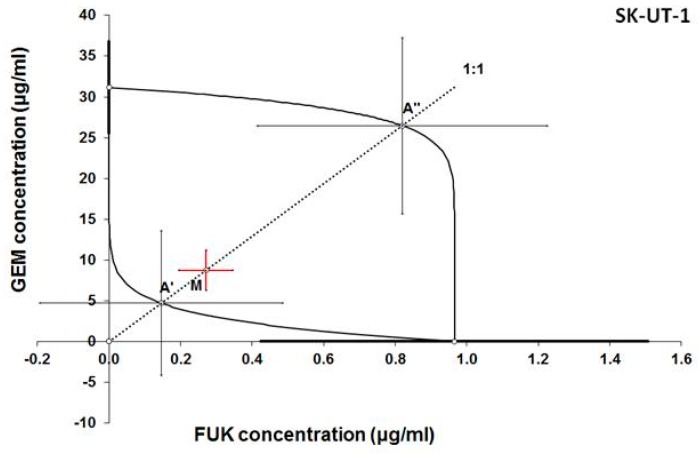
Isobologram showing interaction between gemcitabine (GEM) and fucoidan (FUK) with respect to their anti-proliferative effects in the cancer cell line (SK-UT-1) measured in vitro by the MTT assay. The experimentally-derived IC_50 mix_ value is placed within the area of additivity and indicates additive interaction between GEM and FUK in this cancer cell line.

**Figure 3 cancers-12-00107-f003:**
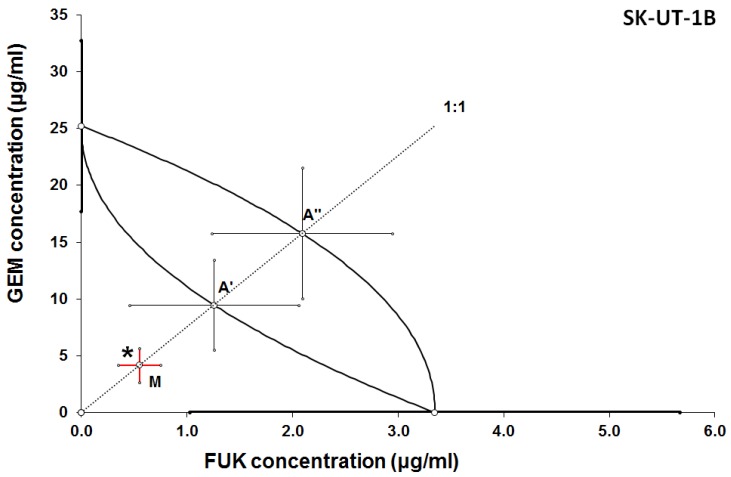
Isobologram showing interaction between gemcitabine (GEM) and fucoidan (FUK) with respect to their anti-proliferative effects in the cancer cell line (SK-UT-1B) measured in vitro by the MTT assay. Because the experimentally-derived IC_50 mix_ value is placed significantly below the point A’, the interaction between GEM and FUK for the cancer cell line SK-UT-1B is supra-additive (synergistic). * *p* < 0.05 vs. the respective IC_50_
_add_ values.

**Figure 4 cancers-12-00107-f004:**
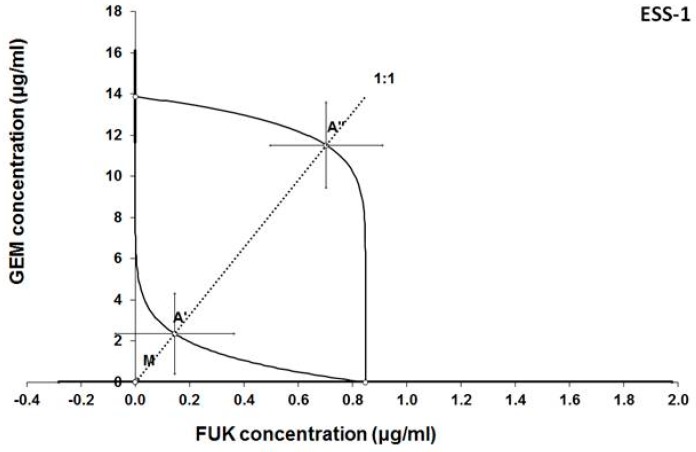
Isobologram showing interaction between gemcitabine (GEM) and fucoidan (FUK) with respect to their anti-proliferative effects in the cancer cell line (ESS-1) measured in vitro by the MTT assay. Although the experimentally-derived IC_50 mix_ value is placed below, but near to the point A’, the interaction between GEM and FUK in this cancer cell line is additive.

**Figure 5 cancers-12-00107-f005:**
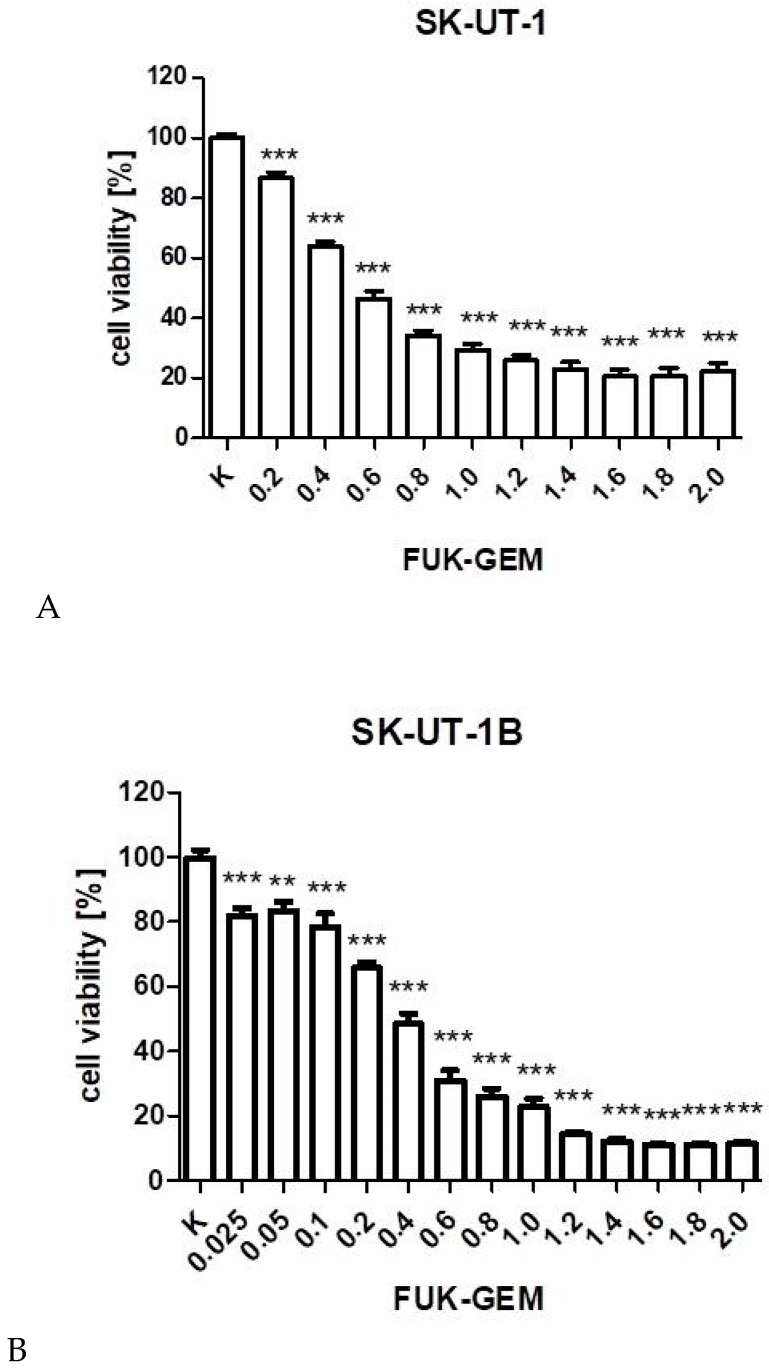
The influence of combined fucoidan and gemcitabine on the proliferation of carcinosarcoma cell lines (SK-UT-1 (**A**), SK-UT-1B (**B**)) and endometrial stromal sarcoma cell line (ESS-1 (**C**)). The cells were treated with fucoidan and gemcitabine at various concentrations for 96 h. (** *p* < 0.01, *** *p* < 0.001 were considered as statistically significant). The values on axis X represent the multiplicity of calculated IC_50_. The combinations of gemcitabine and fucoidan were mixed 1:1 before added to cells.

**Figure 6 cancers-12-00107-f006:**
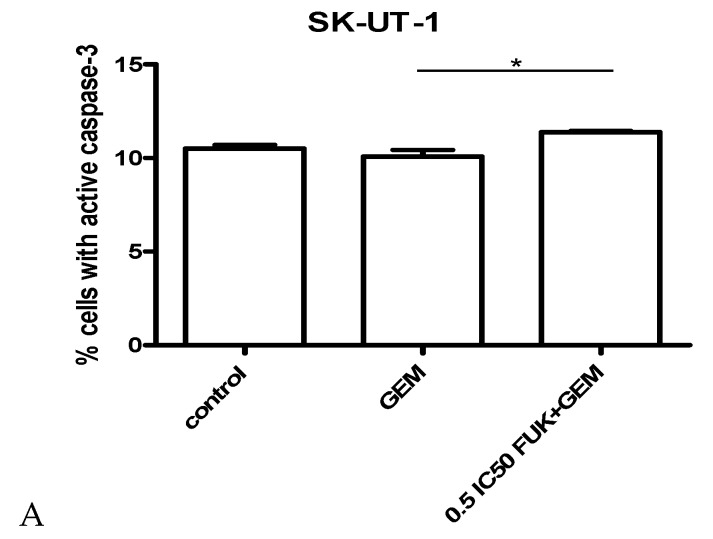
Effects of gemcitabine (in IC_50_ concentration) and mixture of fucoidan and gemcitabine (in concentrations of 0.5 IC_50_ for SK-UT-1 and SK-UT-1B and 0,05 IC50 for ESS-1) on caspase-3 activation in SK-UT-1 (**A**), SK-UT-1B (**B**), and ESS-1 (**C**) cells. Results are expressed as mean ± SD of three separate experiments (* *p* < 0.05, ** *p* < 0.01, *** *p* < 0.001 versus the control, one-way ANOVA test).

**Figure 7 cancers-12-00107-f007:**
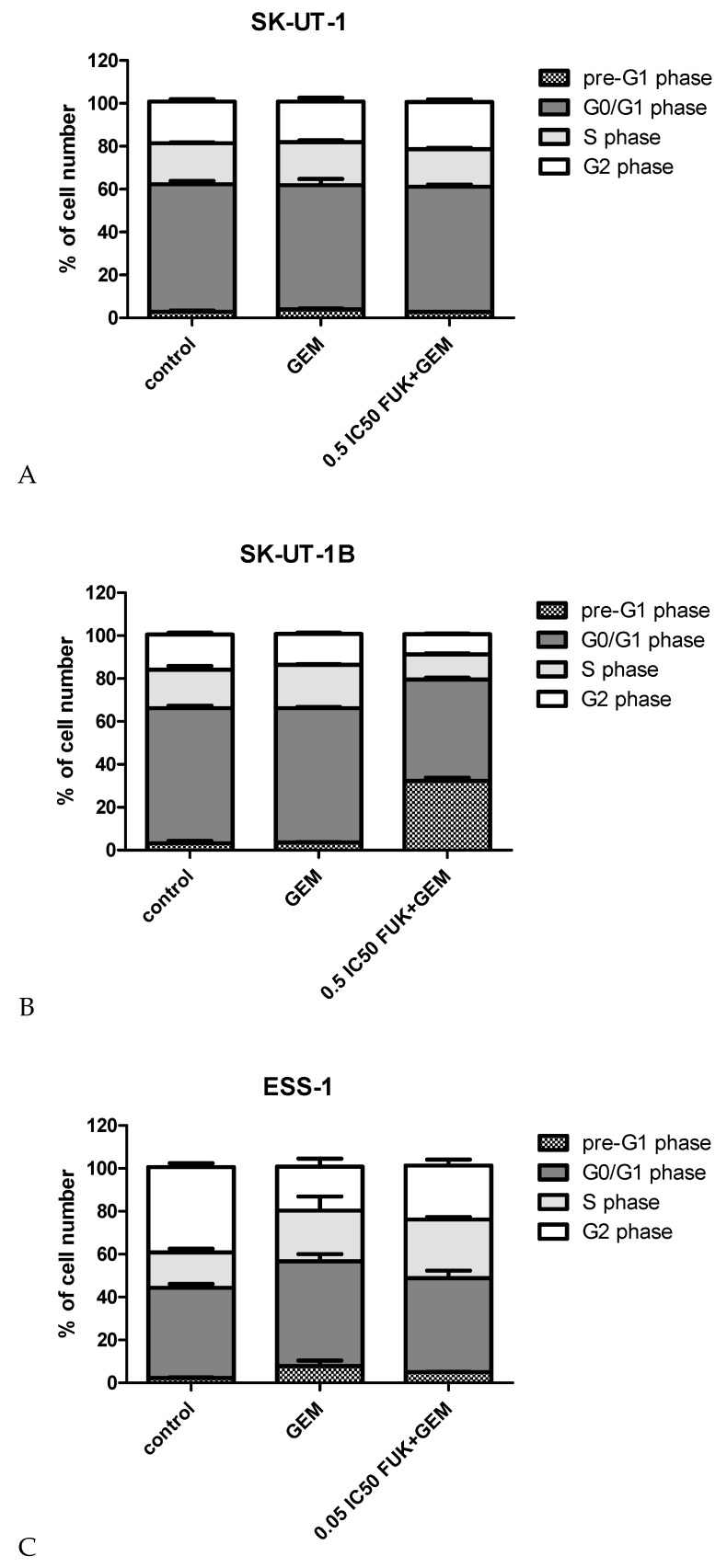
Effects of gemcitabine (in IC_50_ concentration) and mixture of fucoidan and gemcitabine (in concentrations of 0.5 IC_50_ for SK-UT-1 and SK-UT-1B and 0.05 IC_50_ for ESS-1) on cell cycle progression in SK-UT-1 (**A**), SK-UT1-B (**B**), and ESS-1 (**C**) cell lines. The results are presented as mean ± SD from three separate experiments.

**Table 1 cancers-12-00107-t001:** The characteristics of used cell lines [14,15].

Cell line:	SKUT-1 *	SKUT-1B *	MES-SA	ESS-1
**Organism:**	Homo sapiens, human	Homo sapiens, human	Homo sapiens, human	Homo sapiens, human
**Tissue:**	uterus	uterus/endometrium	uterus	uterus
**Culture properties:**	adherent	adherent	adherent	adherent
**Disease:**	grade III, mesodermal tumor (mixed); consistent with leiomyosarcoma	grade III, mesodermal tumor (mixed); consistent with leiomyosarcoma	grade III, recurrent, uterine leiomyosarcoma [14]	endometrial stromal sarcoma
**Age:**	75 years	75 years	56 years	76 years
**Gender:**	female	female	female	female
**Ethnicity:**	Caucasian	Caucasian	Caucasian	Caucasian

* SKUT-1 and SKUT-1B cell lines were derived from the same patient from different sites of tumor SKUT-1 line is a model of sarcomatous part of the tumor (forms spindle cell sarcomas), SKUT-1B is a model of carcinomatous part of the tumor (forms well differentiated adenocarcinomas).

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
