# Peer review of "Isobolographic Analysis Demonstrates the Additive and Synergistic Effects of Gemcitabine Combined with Fucoidan in Uterine Sarcomas and Carcinosarcoma Cells"

_cancers, 2019, doi:10.3390/cancers12010107_

Round 1
Reviewer 1 Report
This manuscript applied isobologram to study the additive effect/synergistic effects of fucoidan and gemcitabine on uterine sarcomas and carcinosarcoma. Authors observed additive/synergistic effects on three cell lines, not MEES-SA, treated with gemcitabine and fucoidan. They concluded that better or at least equal effects of combination of these two drugs on uterine carcinoma/carcinosarcoma compared to monotherapies.
In my opinion, these results or evidences is too weak to support the treatment in uterine sarcomas and carcinosaroma cells. Only one method has been applied in study. No clinical samples have been used. More methods and designs are essential for these two drugs used in combination for uterine sarcomas and carcinosarcoma cell.
Other comments:
Major:
There are four stages for uterine sarcoma. To what stage of these cell lines belong?
Both of these two drugs decrease the cell viability, showed in the “Results”. However, only the cell viability and isobolographic analysis are not enough to get such conclusion. More evidences, such as, flow cytometry, etc, are needed.
Minor:
Fig. 5.: Are the concentrations listed below X-axis is for Fuk-Gem mixture? How is mixture mixed? Line 47/63: i.a? “Abstract” , Line 30-34: should be rewrite.
Author Response
RESPONSE TO REVIEWER 1
Dear Reviewer,
We are very grateful for your comments. We revised the manuscript in accordance with your advice. All the corrections and changes have been marked in yellow. Below you can find a one-by-one response to your comments.
Detailed description of used cell lines was included in the manuscript (table 1) uterine leiomyosarcoma MES-SA cell line was derived from recurrent tumour, all the details You will find in table mentioned below. We provided information from ATCC database and original manuscript describing the cell line [Harker WG, MacKintosh FR, Sikic BI. Development and characterization of a human sarcoma cell line, MES-SA, sensitive to multiple drugs. Cancer Res. 1983 Oct;43(10):4943-50] According to Your suggestions we performed further experiments and added two more methods: caspase 9 apoptosis test and cell cycle assessment with flow cytometry. Appropriate information was included in the manuscript. In legend of figure 5. appropriate explanation was added English changes were performed including “i.a." Lines 30-34 were rewritten
We hope that corrected manuscript will meet Your expectations.
Reviewer 2 Report
The manuscript by Bobinski et al 2019 examines the potential for beneficial co-administration of gemcitabine and fucoidan. The development of novel drug combinations may certainly improve the efficacy of chemotherapy, therefore the subject matter is of interest and suitable for publication.
However, I am concerned about potentially a major flaw in study design and that is that the authors have chosen to only incubate cells with these agents for a time point of 96 hours (4 days). In my experience of cell culture work, this is not only unusual but will likely lead to cells that reach confluence prior to the 96 hours, and deplete media agents such that the cells are stressed and in some cases dying or dead. Furthermore, an incubation period of such length bears no resemblance to cycles of chemotherapy experienced by patients. It would be more typical for toxicity studies to be undertaken at say 3, 6, 12, 24 and possibly 48 hours, but certainly not 96 hours. To provide more of a relevant insight into potentiation or synergistic effects between these agents, the authors should perform experiments at other time points. Additionally, the (control) cells should be monitored and checked that they are both healthy at each of the time points, and still proliferating, the latter of which could be checked via a BrdU incorporation assay. Furthermore, MTT assays are not able to directly distinguish between agents with cytostatic properties as opposed to those that are truly cytotoxic (death inducing), hence further experimental validation with say LDH assays is required to support the data.
Additionally, there are a number of other minor concerns with the manuscript:
It would also be useful for the reader for the authors to compare the results directly with other studies and therefore convert the units from say ng/ml for gemcitabine to molarity as well. Concentrations of agents currently used in patient practise should also be detailed and referenced. The graphs such as Figure 1 could be better visualised if plotted by non-linear regression in Prism, with IC50s shown on the graph. Provide further details about the cell lines and why these were specifically chosen for the assays. Comment on the potential active agent(s) in fucoidan responsible for the effects as well as the proposed mode of action of gemcitabine, and hence the rationale for additive/potentiation/synergistic effects. Typographical and grammatical errors throughout need to be corrected.Author Response
RESPONSE TO REVIEWER 2
Dear Reviewer,
We are very grateful for your comments. We revised the manuscript in accordance with your
advice. All the corrections and changes have been marked in yellow. Below you can find a one-
by-one response to your comments.
According to Your suggestions we added two more methods including apoptosis assessment and cell cycle analysis in flow cytometry. (the methodology is described in the manuscript) We would be glad if we could perform the cell viability assessment as You suggested, but due to the fact that our founding for this project is no longer available we are unable to perform assessments in other time points according Your suggestion. Although I would like to explain that time point of 96h that was selected for our experiments was adopted after careful analysis of methodology published in prestigious journals [1-5] The molarity of gemcitabine was included in material and methods section. Due to the fact that fucoidan is a mixture of sulfated polysaccharides so its molarity may vary depending on fraction. In order not to confuse readers in mixing units we propose to keep ng/ml or mg/ml in the manuscript. We hope You will understand our point of view. Detailed description of used cell lines was included in the manuscript (table 1). We provided information from ATCC database and original manuscript describing the cell line [6] Also justification of cell lines selection was included. The mechanism of gemcitabine action is explained in introduction, in manuscript we added a comment that we discussed the fucoidan activity in detail in previous papers (both are cited in the manuscript) [1,7] Extensive language editing was performed.We hope You will understand the reasons why some of Your suggestions were not included in the manuscript. We hope that corrected manuscript will meet Your expectations and fulfil the requirements of the journal.
References:
Bobiński M, Okła K, Bednarek W et al. The Effect of Fucoidan, a Potential New, Natural, Anti-Neoplastic Agent on Uterine Sarcomas and Carcinosarcoma Cell Lines: ENITEC Collaborative Study. Arch Immunol Ther Exp. 2019 Apr;67(2):125-131. Gumbarewicz E et al. Isobolographic analysis demonstrates additive effect of cisplatin and HDIs combined treatment augmenting their anti-cancer activity in lung cancer cell lines Am J Cancer Res. 2016; 6(12): 2831–2845. Wawruszak et al. Assessment of Interactions between Cisplatin and Two Histone Deacetylase Inhibitors in MCF7, T47D and MDA-MB-231 Human Breast Cancer Cell Lines – An Isobolographic Analysis PLoS One. 2015; 10(11): e0143013. Chen JF et al. STAT3-induced lncRNA HAGLROS overexpression contributes to the malignant progression of gastric cancer cells via mTOR signal-mediated inhibition of autophagy. Mol Cancer. 2018 Jan 12;17(1):6. Deezagi A et al. Prostaglandin F-2α Stimulates The Secretion of Vascular Endothelial Growth Factor and Induces Cell Proliferation and Migration of Adipose Tissue Derived Mesenchymal Stem Cells. Cell J. 2018 Jul;20(2):259-266. Harker WG, MacKintosh FR, Sikic BI. Development and characterization of a human sarcoma cell line, MES-SA, sensitive to multiple drugs. Cancer Res. 1983 Oct;43(10):4943-50 van Weelden G, Bobiński M, Okła K et al. Fucoidan Structure and Activity in Relation to Anti-Cancer Mechanisms. Mar Drugs. 2019 Jan 7;17(1).Reviewer 3 Report
The authors carried out mainly MTT assays to see if cells under the addition of gemcitabine, fucoidan and mixtures would have different viability. Synergistic effect was noticed in SKUT-1B cell line but the interactions could not be observed in MES-SA cell line.
(1) The authors should study the cell death mechanisms promoted by the synergistic effect by examining apoptotic and necrosis pathways, and perform at least one more live/dead assay in conjunction to further strengthen the case.
(2) The authors can consider testing the additive effect in human cancer models given the great disparity between cancer cell lines and human tumor.
(3) The authors can consider a higher dosage for the MES-SA cell line, or adopt another cell line for more conclusive results.
Author Response
RESPONSE TO REVIEWER 3
Dear Reviewer,
We are very grateful for your comments. We revised the manuscript in accordance with your advice. All the corrections and changes have been marked in yellow. Below you can find a one-by-one response to your comments.
According to Your suggestions we added two more methods including apoptosis assessment and cell cycle analysis in flow cytometry. (the methodology is described in the manuscript) We fully understand the limitations of cell lines as a cancer model. Unfortunately the assumptions of this project do not include any other models we could use. On the other hand the advantage of this model is possibility to perform isobolographic analysis to assess the relations between tested drugs in particular cell type. As a continuation of this research we are preparing new project that will include more complex cancer models. We were discussing the problem of dosage of drugs in MES-SA cell line. But in light of our previous research we took into account the effect of tested drugs on normal (benign) cells. We know that gemcitabine in concentration of 100 ng/ml decreases cell viability in human fibroblasts to about 50%, so in higher doses might be toxic. On the other hand concentrations of fucoidan available in human serum are not higher than 1-5 mg/ml. Summarizing above we decided not to increase doses of tested compounds.
We hope that corrected manuscript will meet Your expectations and fulfil the requirements of the journal.
Round 2
Reviewer 1 Report
Although the data showed that cell cycle arrest and apoptosis observed in Figure 7 and supplement Table 3, authors still need to have a solid description and conclusion in Discussion. Description of others conclusions or results in conclusions is not enough. Authors should have their own conclusion about cell cycle and apoptosis, which did not shown in “Conclusion” or “Results”.
English editing is still essential for this manuscript since there are some sentences hard to read and to understand
Author Response
Dear Reviewer,
Thank You again for Your comments, the manuscript was corrected according to them.
In "Discussion" more details regarding results obtained in cell cycle analysis were added.
Paragraph "Conclusions" was updated and partially rewritten.
English editing was performed and spelling errors corrected.
We are open to respond Your further remarks.
We hope that the manuscript in this version will meet Your expectations.
Kind regards.
Marcin Bobiński
Reviewer 2 Report
The authors have made a good attempt at addressing the majority of original concerns. I am happy for the manuscript to proceed to publication.
Author Response
Thank You so much for Your comments and suggestions. We are so glad that the article met Your expectations.
Kind regards.
Marcin Bobiński